# Economic development, weather shocks and child marriage in South Asia: A machine learning approach

Stephan Dietrich[1]*, Aline Meysonnat[2], Francisco Rosales[3], Victor Cebotari[4], Franziska Gassmann[5]

**1** UNU-MERIT and Maastricht University, Maastricht, Netherlands, **2** University of Washington, Daniel J. Evans School of Public Policy and Governance, Seattle, Washington, United States of America, **3** ESAN Graduate School of Business, Lima, Perú, **4** University of Luxembourg, Office of the Vice-rector for Academic Affairs, Esch-sur-Alzette, Luxembourg, **5** UNU-MERIT, Maastricht, Netherlands

* s.dietrich@maastrichtuniversity.nl

**Data Availability Statement:** All data files and scripts are available from the OSF database (osf.io/dqfh8).

## Abstract

Globally, 21 percent of young women are married before their 18th birthday. Despite some progress in addressing child marriage, it remains a widespread practice, in particular in South Asia. While household predictors of child marriage have been studied extensively in the literature, the evidence base on macro-economic factors contributing to child marriage and models that predict where child marriage cases are most likely to occur remains limited. In this paper we aim to fill this gap and explore region-level indicators to predict the persistence of child marriage in four countries in South Asia, namely Bangladesh, India, Nepal and Pakistan. We apply machine learning techniques to child marriage data and develop a prediction model that relies largely on regional and local inputs such as droughts, floods, population growth and nightlight data to model the incidence of child marriages. We find that our gradient boosting model is able to identify a large proportion of the true child marriage cases and correctly classifies 77% of the true marriage cases, with a higher accuracy in Bangladesh (92% of the cases) and a lower accuracy in Nepal (70% of cases). In addition, all countries contain in their top 10 variables for classification nighttime light growth, a shock index of drought over the previous and the last two years and the regional level of education, suggesting that income shocks, regional economic activity and regional education levels play a significant role in predicting child marriage. Given the accuracy of the model to predict child marriage, our model is a valuable tool to support policy design in countries where household-level data remains limited.

## 1. Introduction

Globally, 21 percent of young women are married before their 18th birthday, and South Asia alone is home to over 40 percent of all child brides worldwide [1]. Child marriage constitutes a human rights violation [2], with significant socio-economic consequences for the well-being of children. Fertility rates are higher among child brides. They are more likely to have children

**Funding:** The author(s) received no specific funding for this work.

**Competing interests:** The authors have declared that no competing interests exist.

at an early age, and the risk of maternal and infant mortality is higher [3–5]. Early school drop-out and lower educational attainments further limit the development of child brides [6]. The practice of child marriage is not only detrimental for the individual but for society as a whole. A reduction in child marriage rates was found to increase GDP per capita and lower fertility rates even in the context of sustained population growth [7]. Ending child marriage is therefore associated with substantial social and economic welfare gains, which could sum up to "trillions of dollars between now and 2030" [7].

Over the past decades, the prevalence of child marriage has been decreasing globally [1,8]. South Asia has witnessed the largest decline in the prevalence of child marriage during this period, with a decrease from 49 to 30 percent [1]. Yet, this decline cannot be attributed solely to national programs aiming at reducing child marriage [9]. It implies the existence of a wider range of factors that may play a role. While the factors contributing to child marriage at the household level have been increasingly studied in the literature, there is limited research on the aggregate regional macro-economic factors that influence the prevalence of child marriage (for a broad review of literature on drivers of child marriage, see [8,10]. There is a lack of empirical evidence on whether and how regional determinants influence the prevalence of child marriage. Evidence on models that inform on where child marriage might most likely occur following covariate shocks is even scarcer. This study aims to fill this gap by looking at region-level indicators to predict the persistence of child marriage. We develop a prediction model that relies largely on regional and local inputs to model the incidence of child marriages. The main objective of the paper is to explore the potential of algorithms to support anti-child marriage policy programing. The study also contributes to the literature on child marriage and the relation between aggregate economic development and individual wellbeing in three ways.

First, this study uses machine learning techniques as a tool to predict the changing prevalence of child marriage. Empirical research on child marriage typically attempts to establish causal relationships resting on strong identifying assumptions. While causal inference is important to understand the child marriage phenomenon, in some policy applications it is not central [11]. In the case of child marriage, accurate child marriage forecasts could support policy makers in responding timely and region-specific to changes in local incomes, for instance due to natural disasters or economic recessions, which might have very different impacts depending on the region where they occur. The aim of our prediction model is to forecast child marriage rather than explaining the underlying mechanisms. More specifically, the model aims to help policy makers to allocate anti-child marriage resources to as many girls at risk as possible. A study in India [12] applied machine learning to predict child marriage, however, in contrast to their study we rely on regional predictors and largely avoid individual and household level predictors that could be the consequence of child marriage. A key limitation in most empirical analyses of the dynamics of child marriage is that brides typically join the groom's family after marriage. Therefore, household level predictors such as wealth or age at first sexual intercourse etc. could be the result of marriage which renders them less suitable for child marriage forecasting. We utilize a machine learning model that combines marriage information with remotely sensed data to predict child marriages and simulate the spatial variation in impacts of income shocks. The gradient boosting model we utilize is flexible enough to capture complex interdependencies of predictors, but at the same time it offers ways to assess weights of single predictors avoiding a black box procedure that lacks interpretability and accountability needed for transparent policy processes [13–15]. We show that our model predicts a large portion of child marriages which could make it a relevant tool to support anti-child marriage programming.

Second, while in other studies household surveys have typically been combined with one geo-spatial dataset in isolation, we use several geo-spatial datasets in combination with

household level survey data to analyse the combination of economic forces that predict child marriage. This enables us to consider different sources of regional income changes as predictor of child marriages including natural disasters but also economic recessions. As brides typically join the groom's family after marriage, studies have therefore resorted to the use of geo-spatial data to reflect on the environment in which the bride's family took the decision to marry off their daughter. In this study, we look at the effects of economic development on child marriage by using the nighttime light data (NTL), and at weather shocks by employing Standardized Precipitation Evapotranspiration Index (SPEI) data. For the most part, existing studies show that areas with a lower economic development have higher child marriage rates [16,17]. Fine-grained data sources such as satellite and rainfall data allow to combine household survey data with geo-spatial data and to shed light on different aggregate economic forces that influence socio-economic phenomena. Rainfall data have been used to estimate the consequences of weather shocks at the time of birth on adult health, education and socio-economic outcomes [18], HIV rates [19], child health [20] and educational attainment [21], amongst others. In Sub-Saharan Africa and India, a study employed Demographic Household Surveys and rainfall data to look at the implications of weather shocks on the timing of marriage and found that the age of marriage responds to the short-term changes in the aggregate economic conditions caused by weather shocks, though with differential impact directions depending on regional dowry or bride price practices [22].

Third, the study adds a comparative focus to the analysis of child marriage by unpacking the evidence in four South Asian countries, namely Nepal, Pakistan, Bangladesh, and India. Existing studies often rely on a single country context and this approach limits our understanding of whether the changes in child marriage rates are due to common or country-specific indicators. By utilizing a cross-country approach, this study reveals both common and specific predictors of child marriage in the South Asia region, providing a more solid empirical ground for targeting relevant policy actions. Besides that, we use interpolation techniques to map the prevalence of child marriages in the region and illustrate the spatial variation in the effects of economic shocks on child marriages.

We find that the gradient boosting model that incorporates regional input data, such as droughts, floods, population growth and night light data is able to identify a large proportion of the true child marriage cases. However, the accuracy of predictions varies by country. The model correctly classifies 77% of the true marriage cases, with a higher accuracy in Bangladesh (92% of the cases) and a lower accuracy in Nepal (70% of cases). Given the complexity in the prevalence of child marriage, our model performs remarkably well and predicts a meaningful amount of variation in the prevalence of child marriage in Bangladesh, India, Nepal and Pakistan. Comparing common predictors across countries, we find that changes in regional economic activity as well as negative income shocks and education play a crucial role in the explanation of child marriage prevalence in South Asian contexts. A simulation of the effects of a drought, extremely wet conditions, and a shock in economic growth hints at large effects on the risk of child marriages, however, with considerable spatial variation in the effect size and direction.

The remainder of the paper is structured as follows. Section 2 discusses the predictors of child marriage with a particular focus on aggregate economic determinants of child marriage. Section 3 is devoted to data and methodology. In Section 4 we present the results of the analysis. The paper concludes in Section 5 with the discussion, limitations and possible extensions for future work.

## 2. The determinants of child marriage

This study refers to child marriage and early marriage interchangeably. Child marriage – defined as a legal or customary union before the age of 18 – is a prevalent custom in South Asia, particularly among girls. Even though the incidence in the region has declined over the last decade, one in three young women is still at the risk of being married as a child [17]. While the phenomenon exists for boys [23], globally only 4.5 percent of young men were married before their 18th birthday [24]. Child marriage has been linked to poor socioeconomic, educational, and health outcomes for women [25,26]. Evidence shows that these deprivations are likely to transcend to future generations [27].

The predictors of child marriage are diverse and often interlinked. Both micro- and macrolevel, as well as social and economic factors have been linked to the prevalence of child marriage in different contexts [10,28,29]. At the micro-level, economic considerations such as poverty [5,30–33], and financial considerations related to dowry [16,34–38] have been associated with child marriage. However, child marriage also occurs in wealthier households, as shown in Nepal [39] and other contexts [10]. It implies that other factors, such as gender norms, religion, prestige, and safety associate with socio-economic drivers to explain child marriage rates [29,39–42]. The literature also points at the location as a key determinant of child marriage, with children living in rural areas having a higher risk of child marriage [8]. In particular, rural areas in South Asia lag behind in terms of infrastructure and economic opportunities, which are linked to child marriage [28,33,43]. Children living in rural areas are also deprived of educational opportunities, and their families are more likely to be financially distressed, which may explain the higher incidence of child marriage rates [44]. Other factors related to child marriage include a household's socio-economic status [8] and parental educational attainment [33,45,46]. Overall, in areas with limited employment opportunities, the lack of infrastructure and restricted education services associate with higher child marriage rates [16,17].

The predictors of child marriage go beyond the micro level. Economic, social and political factors at the country or regional level also play a role into how the micro-level drivers associate with child marriage. In South Asia, families used to marry their children early to counteract high infant and maternal mortality rates [47]. Given the improvement in child and maternal wellbeing in the region over the last three decades, this may not be the case anymore. The total fertility rate in South Asia has decreased, from 4.3 children per woman in 1990 to 2.4 children in 2018 [48]. A lower fertility rate was found to associate with a wider range of positive outcomes, including with lower child marriage rates [3–5]. However, this evidence can be argued both ways, in that reducing the prevalence of child marriage may also decrease the total fertility rate, as was recently found in a global study on the impact of child marriage on total fertility [49].

While the effects of demographic factors may vary per country and context, the economic vulnerability of families is perhaps the most consistent factor in explaining the prevalence of child marriage. Countries lagging behind in economic development, as measured by GDP per capita, have the highest child marriage rates [50]. In 2018, of the poorest 25 countries worldwide, 15 had child marriage rates exceeding 30 percent [1]. Using district-level household surveys from 22 Indian states covering 694 districts, a study found that areas with better macroeconomic conditions and lower poverty rates also have lower child marriage rates [51].

Beyond the family level, economic factors that reflect on macro-level dimensions were found to associate with child marriage. Predictors of economic trends, such as average growth in the economic activity of a region are found to associate negatively with child marriage [8]. The expansion of the manufacturing sector, in particular the garment industry, has noticeably increased the female labor market participation in South Asian countries [46–52]. In

Bangladesh it became more common for women to aspire for employment and a career before getting married [53]. The increase in female labor participation also has positive effects on higher educational attainment and reduced fertility rates [54]. Moreover, access to paid employment adds to the women's position in the family, community, and society, and gives them more influence over decisions on marriage [37,55].

Covariate shocks such as economic crises or natural disasters also associate with child marriage rates in South Asia. In India, evidence shows that weather shocks such as droughts influence the age of marriage more negatively [22]. In Indonesia, a study found that families consider child marriage an acceptable coping strategy in the event of an economic shock [56]. Studies show that child marriage is used to deal with the economic consequences in the aftermath of covariate shocks, such as the 2004 Tsunami in India and Sri Lanka, and the floods in Pakistan and Bangladesh [57–61]. Economic development in the region plays a mitigating role, but in the absence of social protection policies that can provide a safety net in the event of covariate shocks, vulnerable households may opt for an early marriage of their children. In these contexts, public health, education, and employment policies are critical for the reduction of child marriage [55,62].

More often than not, child marriage is a strategy to mitigate the risks of an uncertain socioeconomic context. From the perspective of the household, the daughter's future is secured and the household benefits from the bride price [63]. However, the direction of the effect depends on the nature of the local marriage market [22]. In Vietnam, where it is customary to pay a bride price, child marriages can serve as a coping strategy, while in India where dowry is common, parents may delay the marriage of their children because they cannot afford the financial investment [64]. An analysis of a flooding on marriage rates in Pakistan found declining marriage rates for adults, but not for children [65]. Overall, the evidence confirms that the potential costs and benefits of an early marriage are part of a household's decision making during a covariate shock.

The reviewed evidence presented above shows that the harmful practice of child marriage is often the result of poor socio-economic circumstances. A lower socio-economic position and exposure to covariate shocks are often predictors for child marriage. It also implies that economic development can be a mitigating factor for reducing the incidence of child marriage.

## 3. Method

### 3.1 Data

**Demographic and Health Survey (DHS).** We rely on multiple rounds of DHS survey data to analyze the prevalence of child marriage in South Asia. The DHS collects data on household characteristics, health, and nutrition and provides cluster GPS coordinates which allows us to match geo-referenced data to the DHS survey data. Sampling procedures are based on a stratified two-stage cluster design in which enumeration areas are first drawn from census files from which households are randomly selected in the second step. This results in samples representative down to the department and residence (urban-rural) level.

For this analysis we use all survey rounds for which GPS coordinates were available at the time of the study in Nepal (2000/1, 2005/6, 2011/12, 2017), Pakistan (2006/7, 2017/18), Bangladesh (1999/2000, 2004, 2007, 2011, 2014), and India (2015/16). We use multiple survey modules to obtain information on household and individual characteristics as well as to calculate regional demographic, health, and socio-economic indicators.

In the main analysis we focus on the cohort of 20 to 24 years old women and their marital status when they were 15, 16 and 17 years old. Focusing on the age bracket from 20 to 24 is the standard in the child marriage literature as it reduces the risk of misreports of illicit child

marriage practices which tends to be high among women who are underage at the time of the interview [66]. The convention of using the retrospective view covering women aged 20-24 at the time of the interview has been set to avoid missing those women who marry after the interview but still before the age of 18, and to avoid overestimating child-marriage by counting girls aged 15-19 who are married at any given time but are 18 or 19, and thus not defined as children as per the international convention [67]. In contrast, the full retroactive sample of 18- to 49-year-old women who married underage might go too far back in time such that the time of the interview does not adequately represent the macroeconomic conditions under which a girl was married. For example, 49-year-old women interviewed in 1990, were underage until 1968, a time in which nighttime light data and other regional data was not available.

Thus, we analyze whether local conditions predict the marital status of 20- to 24-year-old women when they were underage. As a robustness check we also consider the age bracket 18- to 22-year-old women, which leads to similar results. We set the lower age boundary to 15 to be in line with the cutoff commonly used in DHS women module questionnaires.

The combined data sets include 985.792 women between 15 and 49 out of which 164.070 were of age 20 to 24 at the time of the interview, and that were not married when they were 14 years old. This includes 116.855 women in India, 15.016 in Pakistan, 7.720 in Nepal, and 24.479 in Bangladesh. In the analysis, we consider the marital status of these women when they were 15, 16 and 17 years old and examine if local economic conditions can be used to predict the occurrence of child marriages. We pool the data, meaning that each woman can appear up to three times at age 15, 16, and 17 in the data. As illustrated in Table 1, the year after a girl got married, she drops out of our sample because we are interested in predicting (child) marriage events after which their child marriage status cannot change anymore. In other words, local circumstances can only play a role for marriage events before they occurred. Given the retrospective perspective, we cover child marriage information that spans the years from 1993 to 2015.

**Geo-referenced information.** We complement the DHS survey data with geo-referenced information that approximate the local environment in which child marriages occurred. We use the DHS coordinates of each interview cluster and match them with other geo-referenced data sources. The DHS dataset provides coordinates that are randomly displaced by 0-2 kilometres in urban clusters and 0-5 kilometres in rural clusters with 1% displaced 0-10 kilometres to protect confidentiality of responses [68]. To approximate the local environment in which households reside, we drew a circle with a radius of 50 km around each DHS coordinate. We selected a radius of 50 km to account for the random displacement of DHS coordinates and to capture an area large enough to include the bride's and groom's original households in most cases. In the matching process, we exclude marine areas and foreign territory from these zones, which can decrease the area considered to estimate the geo-referenced variables in

**Table 1. Marital status and age composition.**

| Age = 15 years | Age = 16 years | Age = 17 years |
|---|---|---|
| Married (n= 3977) | | |
| Not-married (n= 160093) | Married (n= 9893) | |
| | Not-married (n= 150200) | Married (n= 12674) |
| | | Not married (n= 137526) |

**Note**: DHS data include 164070 women age 20-24 at the time of the interview that were not married at age 14. The year after a marriage occurred, data points are removed (13870 cases by age 17).

some cases. Based on the zone around each household, we match several proxy indicators for positive and negative income developments and local population densities to the survey data.

We use annual nighttime light data (NTL) from 1992 to 2013 provided by the National Oceanic and Atmospheric Administration (NOOA) to approximate regional level economic activity. NTL has increasingly been used in the literature as an effective approximation for economic activity like GDP or income growth [69–72], income inequality [73] as well as the size of the informal economy [74]. Using NTL is especially suited if official data on economic growth and economic activity over time is poor or not available [70,72,73]. NTL can also help to estimate economic statistics at the supra- and subnational level for which traditional data is more difficult to obtain [72]. An example for this is a study conducted in 2011, which concludes that NTL is a useful tool to better understand GDP at the district level in the case of India [69]. Another study also used a 50 km radius and DHS data to show that NTL indicators approximate local wealth well [75]. However, the indicator is not free of shortcomings. The NOOA NTL values are capped at 63 which limits growth possibilities and information after 2013 is not available. Albeit granular NTL data sources exist [69,75], they are not longitudinal enough for our purposes as they do not overlap with most of the DHS surveys in our data base. In contrast to the DHS data, NTL is available annually and therefore allows for a more realistic approximation of the economic context in which marriage decisions were taken. Besides that, instead of just considering the current level of wealth or economic activity in a province, the data can be used to estimate growth rates over past years to capture economic trends. Based on the NTL data, we matched a range of indicators to the DHS survey data including the average NTL level, growth rate and distribution of NTL in the year and two years before women were 15, 16 and 17 years old within a 50 km radius of each DHS cluster. As changes in the local environment are unlikely to have an immediate effect on child marriages, we match local information in previous years to the DHS data. For example, for a girl that was 16 in the year 2010 we consider the growth in NTL from 2008 to 2009 (and 2007 to 2009 for NTL growth in the past two years). The most recent DHS survey wave (Pakistan 2017/18) includes 1509 women in our sample that were underage in 2015, in which case we can only use the NTL years of the past two years as the NTL data is only available until 2013.

In addition to NTL, we match information on weather extremes to the DHS data to approximate adverse income shocks particularly in agriculturally dependent regions. Here, we rely on high-resolution drought monitoring data for South Asia [76]. The Standardized Precipitation Evapotranspiration Index (SPEI) data is based on bias-corrected precipitation and temperature indicators. The data is available at 0.05˚ spatial resolution since 1981 for South Asia and we refer to [76] for details. There are several ways in which SPEI or climate data more generally could be used to approximate local weather events including annual means, the number of days above or below a certain threshold or peaks in measures [77–79]. We use the latter as we are interested in extreme events and match the minimum and maximum SPEI scores to the survey data using again a radius of 50 km. The minimum score approximates droughts and similar to [80], we use the maximum score as indicator for extremely wet conditions. We first calculated the minimum and maximum annual SPEI score for each year using three months moving averages. Thereafter, we matched the data with the DHS cluster coordinates and calculated variables with the maximum and minimum scores in the year and two years before women were 15, 16 and 17 years old within the 50 km radius of each DHS cluster.

To account for regional population densities, we additionally match population estimates to the DHS data. We rely on WorldPop gridded data for Asia at a resolution of approximately 1 km by 1 km for the years since 2000 in 5-year intervals. The population estimates are based on census data and several ancillary data sources, which improves the accuracy of outdated census data and provides granular spatial population count information [81]. For the analysis

we match the population estimates closest to the years in which women were 15 to 17 within 50 km of each DHS cluster to the survey data.

**Data limitations.** A key limitation in most empirical analyses of the dynamics of child marriage based on secondary data is that brides in virilocal societies typically join the groom's family after marriage [82]. That implies that characteristics of married girls covered in household surveys may not reflect their family background which could have led to the marriage in the first place. Information on non-married women in the cluster of origin of the brides might not give an accurate picture of the characteristics of households that led to the early marriage of girls. By adding geo-referenced information, we seek to capture regional income dynamics that cover the environment in which the bride's family took the decision to marry off their daughter. This holds to the extent that the bride's and groom's family reside in the same region. This is a key assumption necessary for causal inference analyses in this context [22], but may also limit a model's ability to predict the occurrence of child marriages. In contrast to studies aiming to draw causal inference, however, the performance in prediction models can be validated by the accuracy in which the available inputs predict child marriages.

Reporting bias is another shortcoming that most empirical research on child marriage suffers from. As child marriages are illicit in most of the research region, it may lead households to not report marriages of underage household members. Comparing regional marriage reports at the time of the interview with the results on the same region and time using a retrospective analysis (based on later survey waves), hints at considerable underreporting of child marriage in surveys [66]. Therefore, we use a retrospective perspective by focusing on women of age 20 to 24 at the time of the interview, which is also how DHS presents child marriage rates in their reports. Our models are thus trained with data of women that stayed in the same region and did not migrate. By including regional information on the share of household members that are "away" we seek to approximate regions with higher levels of migration in the data. For some survey waves in Bangladesh and Nepal, there is more specific information on whether any household members migrated, which suggests regional migration rates of 4% and 1.6% respectively. By using the age bracket from 18- to 22-year-old women as robustness check, we further corroborate our models and do not find significant changes in aggregate performance measures (see S1 Table).

## 3.2 Measures

**Child marriage in South Asia.** The prevalence of child marriage has decreased over the last two decades on average but remains remarkably high in the SA region. Fig 1 depicts the development of the share of girls in our sample that married in the past year in the whole sample (left panel) and separated by country (right panel). Our data show the staggeringly high rates of child marriage in Bangladesh and Nepal in the 90ies and a steady decline thereafter. The decline in child marriage rates in Bangladesh has been associated with increasing government and non-government awareness campaigns around the issue and increasing education rates and employment opportunities for women [83,84]. Yet, the decline in Bangladesh has slowed in the recent decade. About 5% of girls married in the past year and we observe slight increases in marriage rates in Nepal, India, and Pakistan since 2010 (see S1 Fig for results among 18-22 old women).

To depict regional differences and the spatial variation of the prevalence of child marriages we use the latest DHS wave of each country, the DHS cluster coordinates and spatial interpolation techniques to build prevalence surface maps. In particular, we use the DHS information to estimate the rates of child marriage in the past year for each point on the map where child marriage information is not available. The spatial interpolation hinges on the assumption that

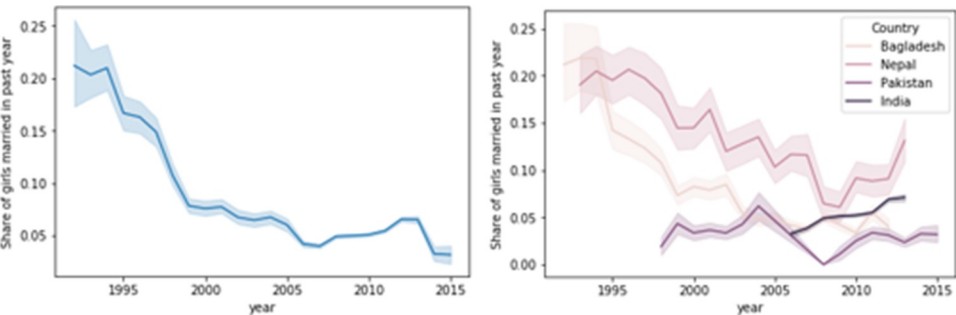

**Fig 1.** Share of girls 15-17 that got married in the past year in whole data (left) and by country (right). ***Notes****: Own calculations are based on women age 20 to24 at the time of interview. Outlier observations of women that were interviewed outside the standard survey period (i.e., women that were interviewed in later or earlier years than the rest of same survey wave) were removed to avoid plotting means that are based on few observations. The shaded area shows 95% confidence bands.*

the variation in child marriages is spatially continuous and that neighboring points are similar. However, the number of observations per survey cluster is low and the observed prevalence and spatial variation may reflect random sampling variation. To account for this, we use the methodological approach and R package described in [85]. Thereby, the smoothed child marriage prevalence for each cluster is calculated from observations located within a circle of varying radius that comprises at least a predefined minimum number of DHS observations. The smoothed child marriage prevalence is then spatially interpolated with adaptive kernel estimators in which the smoothing bandwidth is proportional to the radius of the circle to be drawn around the cluster to capture the minimum number of observations (for details of the approach, we refer to [85]).

Fig 2 shows the interpolated rate of child marriage in the past year when women in our sample were 15 to 17 years old. On average, 6% married in the previous year in the sample, yet we observe large spatial differences. The child marriage rate was highest in the North-East of the research region, and Bangladesh in particular, with an annual marriage rate of up to 12%. In other parts such as the North and South of India, annual child marriage rates were lower.

**SPEI.** South Asia faced multiple severe extreme weather events during the research period. Fig 3 shows the distribution of the minimum and maximum SPEI values in our sample. The average of the minimum SPEI score is -1.5 and about 13% of observations have a minimum SPEI below -2 which is commonly used to define extreme droughts. India shows the largest prevalence of droughts (and the lowest score with -6 in 2010) followed by Bangladesh and Nepal.

Positive extremes of the SPEI score, our proxy for extremely wet conditions, reach a mean of 1.9 where the highest scores were registered for India and Nepal with scores above 3. About 38% of women in our sample resided in regions that experienced extremely wet conditions -characterized by a SPEI value above 2. However, there are considerable differences between countries with India having the highest share in extremely wet conditions (54%) and Pakistan the lowest share (4%) (Bangladesh and Nepal with 16% and 19%, respectively). An example of the spatial distribution of SPEI in the research region is presented in S2 Fig.

**Nighttime lights.** In Fig 4 we show the distribution of mean NTL and NTL growth. The data suggest that NTL tends to be low with a mean below 20 on a scale from 0 to 63 in most cluster areas. On average, regional NTLs have grown by 3.8% but with a large spatial variation. NTL growth was largest in India with 5% followed by Bangladesh with 2% and lowest in Nepal with 1%. However, as the frequency of survey rounds differs by country it should be noted that

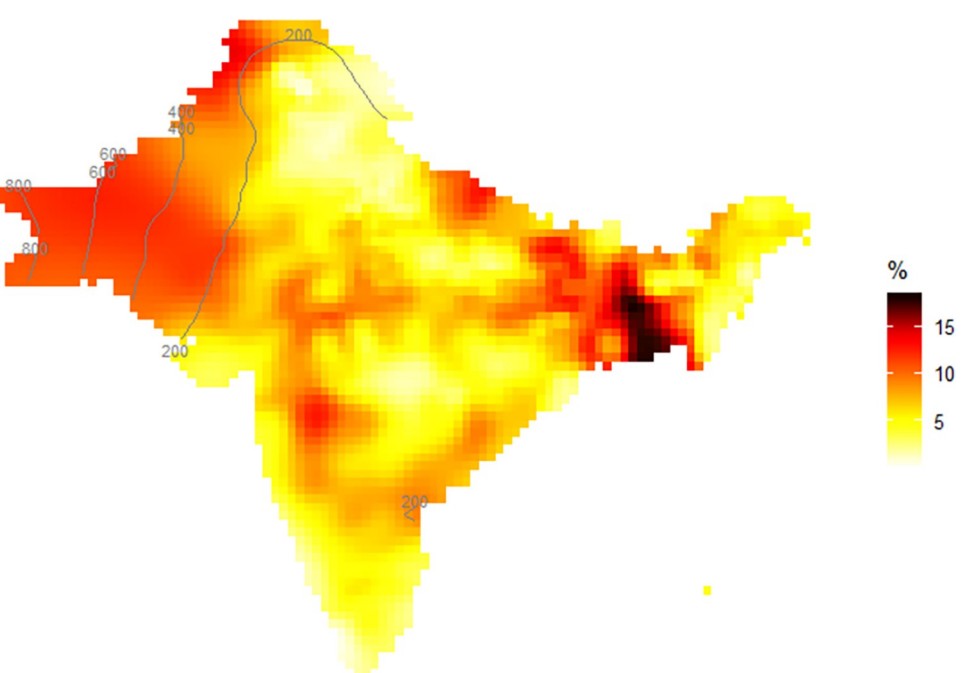

**Fig 2. Prevalence surface of child marriages in the past year in the most recent DHS survey wave. Note**: Based on DHS India 15/16, Nepal 17, Bangladesh 14, Pakistan 17/18. Interpolation based on adaptive Kernel estimator following the methodology of [85]. Grey lines depict smoothing circle radius in kilometers. Adaptive bandwidths set to number of persons surveyed=2256.

we are looking at different time periods in each country. For India, the period covers 2007 to 2013 while for Nepal it covers the years 1993 through 2013. The spatial distribution of NTL in the research region in the year 2013 is presented in S3 Fig.

**Summary statistics.** As women normally leave their parental households after marriage, household information of married women in the DHS data possibly characterize the groom's household but not the original household of the bride and could thus be the consequence rather than the cause of child marriage. Therefore, most of the household characteristics might

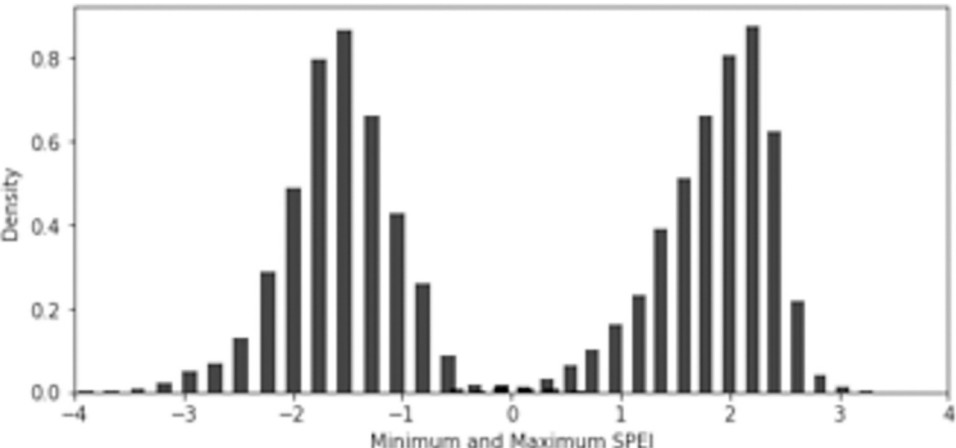

**Fig 3.** Distribution of regional minimum (left) and maximum (right) SPEI scores. *Note: Own calculations based on SPEI data from [76].*

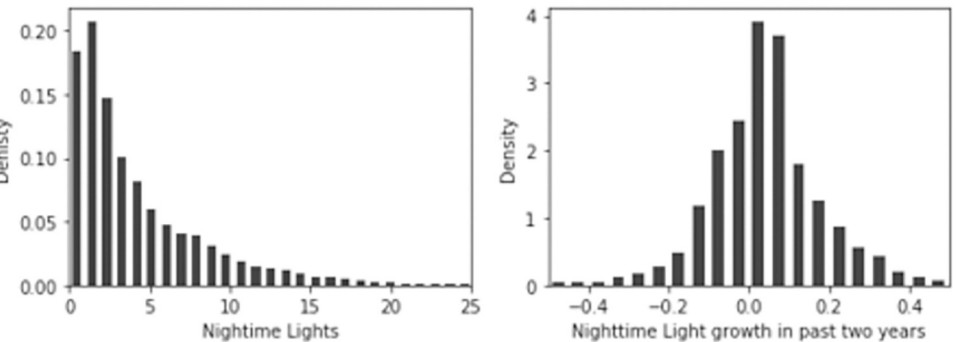

**Fig 4. Distribution of local night-time light and growth. Note**: Own calculations based on NOAA data and 50km radius around each DHS cluster. Larger values reflect more illuminated areas with a possible range of 0-63.

not be useful for predicting child marriages. The only individual predictor we use in the models besides the age of the woman is the years of schooling. We restrict our proxy for education to the age before 15 to ensure that it only covers the period before girls got possibly married. In addition, we generated several regional-level predictors that we summarize in Table 2, including the share of women between 15 to 49 that got married underage, a range of demographic indicators such as the mean age of childbearing, fertility and child mortality rates, indicators of the regional age structure, and socio-economic indicators such as the share of women in employed work or women's mean educational attainment in the region. We define

**Table 2. Summary statistics of district level predictors.**

| Variable | Mean | SD | Max | Min |
|---|---|---|---|---|
| Education in single years (regional) | 4.94 | 1.22 | 8.40 | 1.63 |
| Women Child Marriage under 18 (regional) | 47.03 | 15.11 | 86.44 | 15.74 |
| Household has a radio (regional) | 0.12 | 0.12 | 0.70 | 0.02 |
| Household has a telephone (regional) | 0.05 | 0.08 | 0.51 | 0.00 |
| Household has a television (regional) | 0.61 | 0.21 | 0.97 | 0.06 |
| Currently divorced/separated /widowed (regional) | 0.05 | 0.01 | 0.07 | 0.00 |
| Total Fertility Rate (regional) | 2.44 | 0.69 | 4.69 | 1.17 |
| Mean Age of Childbearing (regional) | 25.68 | 1.56 | 31.35 | 23.11 |
| Currently not married (regional) | 0.20 | 0.03 | 0.31 | 0.10 |
| Dependent (regional) | 0.36 | 0.05 | 0.48 | 0.25 |
| Under-5 mortality rate (regional) | 53.22 | 21.42 | 148.43 | 7.05 |
| Child mortality (regional) | 10.35 | 5.57 | 41.29 | 0.00 |
| Infant mortality rate (regional) | 43.40 | 17.10 | 111.75 | 5.59 |
| Post-neonatal mortality (regional) | 12.25 | 5.84 | 54.57 | 0.00 |
| Neonatal mortality (regional) | 31.14 | 12.26 | 72.47 | 4.40 |
| Currently working (regional) | 0.46 | 0.11 | 0.98 | 0.07 |
| Women employment status (regional) | 0.25 | 0.12 | 0.98 | 0.04 |
| Household has son or daughter elsewhere (regional) | 0.12 | 0.03 | 0.24 | 0.04 |
| Age difference between husband and wife (regional) | 5.68 | 1.85 | 11.38 | 3.38 |
| Women school attainment in single years (regional) | 4.37 | 1.20 | 8.29 | 0.87 |

**Note:** Calculations based on 474363 data points in the analysis sample (15- to 17-year-old women that were 20-24 at the time of the interview) using DHS sampling weights. SD=standard deviation.

regions at the admin 1 level, the lowest level at which the DHS data is representative in all considered countries.

## 3.3 Model

We aim to develop a model that relies largely on regional and local inputs to predict the occurrence of child marriages. Except for a girl's age and education, the model requires no individual or household level input variables, so that it could be used for child marriage forecasting without having to collect costly individual data. In particular, we rely on the geo-referenced proxy indicators for changes in local economic activity and income shocks to predict child marriages. For the modelling we assume that policy makers would be more concerned with detecting true child marriage cases (true positives) than the risk of wrongly classifying girls at risk of child marriage (false positives). In other words, missing a true positive is more costly than the consequences of a false positive. Many current (anti-) child marriage programs are not targeted in a narrower sense and "treatments" are assigned widely, i.e., the vast majority of "treated" girls will not get married underage. We aim to train an algorithm that ensures that the large group of "treated" girls contains as many true cases as possible. Therefore, we would like the model to perform well in capturing true child marriage cases, even if that means that we overpredict child marriage rates. Another important feature of the setting is the imbalance in positive cases. While child marriage rates are shockingly high, there is still a stark imbalance with many more cases of girls that did not get married underage in the previous year. For example, with child marriage rates at 5%, a naïve model that always predicts that there is no child marriage would be correct in 95% of the cases. We account for this imbalance in the modelling by using the ratio of the number of negative class (no child marriage cases) to the positive class (child marriage cases) to reweight the data. We tested several models, out of which tree-based models performed best for this application. The results of the starting point, a logistic regression, are presented in S2 Table. In the following we first describe the model and thereafter discuss the model performance and prediction results. For the modelling, we randomly split the data into a training sample (80%) to train the models and a test sample (20%) used to validate the predictions.

**Gradient boosting trees.** Decision trees segment the data based on shared characteristics into sub-groups that are as similar as possible in the target variable – child marriage in our case. While decision trees fit the data at hand well, out-of-sample prediction accuracy tends to be poor [86]. To improve predictions, many tree-based methods combine several weak learners to a strong ensemble that is less prone to overfitting. Boosting models are very popular in the machine learning community since the presentation of the AdaBoost algorithm in the mid-nineties [87]. For this application we have utilized the XGBoost implementation, developed by [88], which is an open-source public library that provides an "off-the-shelf procedure" for the general gradient boosting framework implemented in different computer languages. However, we will center our discussion on the binary classification problem.

At a higher level, gradient boosting trees possess many of the qualities of classification trees, such as natural handling of data of "mixed type", handling of missing values, robustness of outliers in input space, insensitive to monotone transformations in inputs, computational scalability, and the ability to deal with irrelevant inputs, while fixing some of its main problems, such as lack of interpretability, and low predictive power. The way this is achieved is by assembling different cellular tree models and solving iteratively for a loss function that follows a greedy algorithm that searches for the direction with the steepest descent in such function.

**Selection of hyper-parameters.** Fitting gradient boosting trees to real data requires the selection of various hyper-parameters, e.g., tree depth, or learning rate. A common alternative

**Table 3. Hyper-parameter search grid.**

| Parameters | Description | Minimum | Maximum | Step Size |
|---|---|---|---|---|
| max_depth | Maximum depth of a tree | 3.00 | 18.00 | 1.00 |
| Gamma | Minimum loss reduction required to make a further partition on a leaf node of the tree. | 1.00 | 9.00 | 0.01 |
| reg_alpha | L1 regularization term on weights. | 40.00 | 180.00 | 1.00 |
| reg_lambda | L2 regularization term on weights. | 0.00 | 1.00 | 0.01 |
| colsample_bytree | Subsample ratio of columns when constructing each tree. | 0.50 | 1.00 | 0.01 |
| min_child_weight | Minimum sum of instance weight (hessian) needed in a child | 0.00 | 10.00 | 1.00 |

to make a data driven selection of such a parameter vector, is to evaluate each possible vector value and evaluate the out-of-sample performance of the resulting model with respect to a certain metric. To get the best model specification for our application, we searched over a range of combinations of hyper-parameter values to find the most suitable combination for our setting. Table 3 shows the grid of hyper-parameters that we used. Depending on the context and model objectives, there are several metrics that can be applied to validate and compare the model performance. In this application, we aim to build a model that can help policy makers to target as many girls at risk of child marriage as possible. As a result, we are less concerned with identifying girls at risk that do not get married underage, which is in line with existing anti-child marriage programming practices. Therefore, the hyper-parameter search aims to maximize *recall* in the main analysis, that is the ratio of true positive cases over the sum of true positive and false negative cases:

$$Recall = \frac{True\ Positives}{True\ Positives + False\ Negatives}$$

In practical terms, *recall* expresses how many true child marriage cases are captured by the model predictions. As alternative, we also follow [89] in using the area under the precision-recall curve as performance metric, which leads to very similar results (S4 Table). Policy makers wishing to put more weight on the precision of predictions could increase the threshold at which a case is classified as child marriage. In S4 Fig we present the precision-recall curve that shows the trade-off between precision and recall.

We built a model for the entire data set and one for each country separately. The hyper-parameter values that we obtained after a random grid search procedure comparing 10.000 models with different hyper-parameter combinations are depicted in Table 4. In the models we included all variables described in the data section as well as country and year indicators and latitude and longitude measures to account for location-specific influences. To reduce

**Table 4. Hyper-parameter values.**

| Parameters | Bangladesh | Nepal | Pakistan | India | All Countries |
|---|---|---|---|---|---|
| max_depth | 14 | 6 | 16 | 3 | 16 |
| Gamma | 6.54 | 7.89 | 5.07 | 6.82 | 4.25 |
| reg_alpha | 51 | 156 | 50 | 113 | 175 |
| reg_lambda | 0.64 | 0.08 | 0.88 | 0.88 | 0.68 |
| colsample_bytree | 0.84 | 0.79 | 0.86 | 0.77 | 0.88 |
| min_child_weight | 9 | 7 | 10 | 5 | 5 |

**Note**: Selection based on random grid search procedure with 10.000 iterations.

concerns of a spatially correlated noise structure that may result from using spatial data, we additionally used spatially blocked partitioning of the validation data [90] as a robustness check and investigated the resulting error for spatial correlation both of which did not hint at spatially correlated errors.

## 4. Predictions

The results of the models are summarized in Table 5, where we present the counting of all the possible outcomes of our classifier (true negative, false positive, false negative and true positive) for each country and the pooled data. Note that we only use the test data (20% of data) for the predictions which explains the reduction in the absolute number of cases. In addition to the classification results, we also present several evaluation metrics to assess the model performance including the area under the ROC curve (ROC AUC) as an indicator of explanatory power of the models, accuracy (share of correctly classified cases), precision (true positives divided by sum of true and false positives), recall, and F1 (harmonic mean of precision and recall). We stress that recall is the metric that interests us the most, but the other metrics help to provide a more comprehensive picture of the model performance.

As can be observed in Table 5, the model leads to a recall of almost 0.8 for the cases under study. This means that in all cases of the test data we are able to identify slightly short of 80% of the true child marriage cases. The recall rate is highest for Bangladesh (92%) and lowest for Nepal (70%). The accuracy of predictions suggests that the model correctly classifies 74% of the cases which increases to 84% in Bangladesh and drops to 69% in Nepal. Not surprisingly, the precision of the predictions is low as we trained models to capture true child marriage cases at the cost of false positives. Considering the complexity of marriages as well as the fine line between child and non-child marriage, the model performs remarkably well. The model can capture true child marriage cases in most cases and is able to predict a meaningful amount of variation in the prevalence of child marriages in the region. The model performs better in Bangladesh and Pakistan compared to India and Nepal, but differences are not substantial. We also compared the current results with the ones obtained when the age bracket is between 18 and 22 years of age (see S2 Table), which led to very similar results. This indicates that changes in recall length have no significant implications for model predictions.

The results imply that regional and local variables are weighty predictors of child marriage. In the next step, we take a closer look at the importance of single predictors. The feature

**Table 5. Summary of results.**

| Results | Bangladesh | Nepal | Pakistan | India | All Countries |
|---|---|---|---|---|---|
| **Panel A: confusion matrix** | | | | | |
| True Negative | 10619 | 2045 | 4436 | 44164 | 62623 |
| False Positive | 2148 | 946 | 652 | 19640 | 22024 |
| False Negative | 57 | 124 | 47 | 826 | 1146 |
| True Positive | 683 | 289 | 126 | 2772 | 3779 |
| **Panel B: performance metrics** | | | | | |
| ROC AUC | 0.93 | 0.76 | 0.92 | 0.80 | 0.83 |
| Accuracy | 0.84 | 0.69 | 0.87 | 0.70 | 0.74 |
| F1 | 0.38 | 0.35 | 0.26 | 0.21 | 0.25 |
| Precision | 0.24 | 0.23 | 0.16 | 0.12 | 0.15 |
| Recall | 0.92 | 0.70 | 0.73 | 0.77 | 0.77 |

**Note**: Panel A reports on count of cases in the test data (20% of full sample) and Panel B reports shares.

**Table 6. Predictor importance.**

| Importance | Bangladesh | Nepal | Pakistan | India |
|---|---|---|---|---|
| 1 | drought2 | education | flood2 | education |
| 2 | pop15 | ntlG1 | flood1 | ntlG2 |
| 3 | drought1 | flood2 | drought2 | drought2 |
| 4 | flood2 | drought1 | drought1 | flood2 |
| 5 | flood1 | Pop16 | education | flood1 |
| 6 | education | stdntl15 | ntlG1 | stdntl16 |
| 7 | ntlG1 | drought2 | pop15 | ntlG1 |
| 8 | ntl15 | stdntl17 | pop16 | age |
| 9 | ntlG2 | age | age | ntl17 |
| 10 | ntl17 | ntl15 | ntlG2 | drought1 |

**Note**: Importance of 1 refers to the most important predictor. ntl=nighttime light; stdntl= standard deviation of NTL; pop=population; 15, 16 and 17 refer to levels of variable at age 15, 16 and 17 of girls. 1 and 2 refer to values in the past year and past two years year under consideration.

importance within the classification was extracted for each country and the results are presented independently in Table 6. This scoring type counts how many times a variable is used in the trees for splitting purposes and thus is an important indicator for the classification. In Table 6, we show the ten most important predictors of each model, where 1 denotes the most important predictor. Variables that are listed in each of the five models are shaded in grey to highlight common predictors. All countries contain in their top 10 variables for classification nighttime light growth over one year (ntlG1), flood index over the last 2 years (flood2), shock index over the last year (drought1), shock index over the last 2 years (drought2), and the level of education (education). This suggests that changes in regional economic activity as well as negative income shocks and education play a crucial role in our ability to explain the child marriage phenomenon.

Interpreting the effects of single predictors in the model is complicated because of the potentially complex interdependencies with other variables. A variable can be highly important for predictions, but whether the effect is positive or negative may change depending on the level of other predictors. To illustrate how local income shocks are predicted to affect child marriages, we simulate scenarios in which we vary our proxy indicators for local income dynamics and keep all other aspects constant. We separately consider the effect of changes in NTL growth, the drought and flood indicator. Next, we only change values of the previous year and keep the values of the respective predictors two years before constant.

We first predict a benchmark scenario in which we set the respective income proxy to the regional mean. Thereafter, we change the regional mean of the income proxy by two standard deviations (SD) and examine how it changes child marriage predictions compared to the benchmark scenario. We consider the case of reductions in NTL growth in the past year (as proxy for an economic recessions), a reduction in minimum SPEI score (as proxy for drought) and an increase in maximum SPEI score (as proxy for extremely wet conditions). We present the percentage point (pp) changes in the probability of child marriage graphically using maps based on the same interpolation method as described earlier to illustrate the spatial variation in predictions. To avoid overlaps between data collected in different survey waves, we only present the predictions of the latest DHS survey collected in each country. The results are depicted in Fig 5 with the prediction for a NTL growth shock in the left panel, a drought in the middle, and extremely wet conditions in the right panel. Note that we depict changes in the

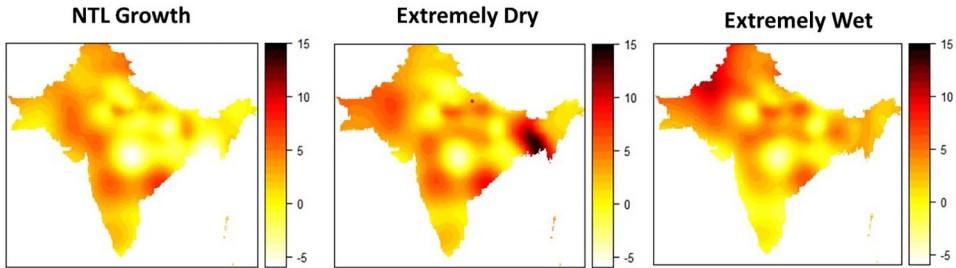

**Fig 5. Predicted effect of local income shocks on child marriage rates of girls 15-17. Note**: Difference in probability of child marriage when NTL growth, drought and flood indicators are changed by 2 standard deviation from district mean holding all else constant/ Based on test data (20%) of DHS India 15/16, Nepal 17, Bangladesh 14, Pakistan 17/18. Interpolation based on adaptive Kernel estimator following the methodology of Larmarange and colleagues [85]. Adaptive bandwidths set to number of persons surveyed=4000.

probability of child marriage and not changes in child marriage classifications as it provides a more nuanced measure of the effects.

Among women aged 20 to 24 in the most recent DHS surveys, our model predicts a child marriage rate of slightly above 5% in the benchmark scenario. Compared to that, our measure for a drought (2 SD reduction in the minimum SPEI) has the largest effect on child marriages. On average, it increases child marriages by approximately 4.6 pp. Yet, there is considerable variation with the largest effects in Bangladesh and coastal Andhra Pradesh with increments of more than 10 pp and slight reductions in the probability of child marriage in Bengal and central India. On the other extreme, the maximum SPEI, our measure for extremely wet conditions, has a slightly lower effect with 2.9 pp on average. The spatial variation of the effect of extremely wet conditions is like the case of droughts, with the exception of Bangladesh where extremely wet conditions play a less important role for child marriage predictions. Lastly, a regional NTL growth shock increases the rate of child marriage predictions by about 0.8 pp on average. Particularly in the West of the research region, predicted child marriage drops after a growth shock, which is opposite to the predicted effects of a drought. The simulations indicate that the effects of changes in local incomes on child marriages are not uniform and vary by type of shock and can vary markedly depending on the region of occurrence.

## 5. Conclusion

While child marriage rates have decreased over the past decades, it remains a widespread practice, especially in South Asia, where 40 percent of all child brides worldwide live. It is widely acknowledged that early child marriages generate health risks and economic costs for women and societies at large. Ending child marriage is therefore a priority for policy-makers worldwide and has been included as a target under the fifth Sustainable Development Goal. Despite a large literature on micro-economic predictors of child-marriage, research on the aggregate factors that influence the prevalence of child marriage remains limited. In this paper, we develop a prediction model that relies largely on regional and local inputs to model the incidence of child marriages. Our model predicts child marriage incidences with an accuracy of 74%, and more importantly, it captures 77% of true child marriage cases. Given the complexity of child marriages, the model performs notably well, without relying on yearly household level data for predictions. We believe that such a model could support anti-child marriage programming and inform how resources could be targeted to girls with the highest risk of child marriage. As we optimized the model for its ability to detect true child marriage cases, it leads to a considerable amount of overprediction of child marriage cases. Many current anti-child

marriage programs are not targeted in a narrower sense and "treatments" are assigned widely i.e., the vast majority of "treated" girls will not get married underage. In the presented application we train an algorithm that focusses on detecting true child marriages cases ex ante that could help to ensure that the large group of "treated" girls contains as many true child marriage cases as possible. If policy makers prefer to reduce the amount of overprediction, algorithm could be trained to account for these preferences.

Such an algorithm is particularly useful in countries where household-level data is limited or infrequent. We find that for Bangladesh, India, Nepal and Pakistan, all countries contain nighttime light growth, a shock index of drought over the previous and the last two years and the regional level of education in their top 10 variables for classification. This suggests that negative income shocks, the regional economic activity and regional education levels play a significant role in predicting regions with high levels of child marriage.

However, our simulation results point at substantial spatial variation with large regional differences in the effects of income developments and the type of shock on child marriages. We find that compared to our benchmark scenario our measure for a drought has the largest effect on child marriages on average, increasing the risk of child marriages by approximately 4.6 pp. A regional NTL growth shock has the smallest effect increasing the rate of child marriages by about 0.8 pp on average, compared to a benchmark scenario.

In this paper we show that a prediction model that mainly requires publicly available regional input data could be useful for child marriage programming particularly in countries where household-level data remains limited. Refining the model and expanding inputs in future work still offers scope to improve the model performance. Given its accuracy in predicting true child marriage cases, such a model could become a useful tool for policymakers to determining regions and designing interventions that could effectively help curtail child marriage practices.

## Supporting information

**S1 Fig. Share of girls 15-17 married in previous year among 18-22 year old women at interview. Note**: Own calculations based on women age 18 to 22 at the time of interview. Outlier observations of women that were interviewed outside the standard survey period (i.e., women that were interviewed in later or earlier years than the rest of same survey wave) were removed to avoid plotting means that are based on few observations. Shaded area shows 95% confidence bands.
(TIF)

**S2 Fig. Minimum (drought indicator; left) and maximum (extremely wet conditions; right) SPEI, 2012. Note**: Own calculation based on SPEI data provided by Aadhar and Mishra (2017).
(TIF)

**S3 Fig. Nighttime lights 2013. Note**: Own calculation based on NOOA NTL data of 2013.
(PNG)

**S4 Fig. Precision-Recall curve. Note**: Based on hyperparameters as displayed in Table 4.
(PNG)

**S1 Table. Summary of results for the 18-22 Age Bracket.**
(DOCX)

**S2 Table. Logistic regression model.**
(DOCX)

**S3 Table. Results with spatially blocked partitioning.**
(DOCX)

**S4 Table. Summary of results (with area under precision-recall curve as performance metric).**
(DOCX)

## Author Contributions

**Conceptualization:** Stephan Dietrich, Aline Meysonnat, Victor Cebotari, Franziska Gassmann.

**Data curation:** Stephan Dietrich, Aline Meysonnat.

**Formal analysis:** Stephan Dietrich, Francisco Rosales.

**Methodology:** Stephan Dietrich, Francisco Rosales.

**Software:** Stephan Dietrich, Francisco Rosales.

**Visualization:** Stephan Dietrich.

**Writing – original draft:** Stephan Dietrich, Aline Meysonnat, Francisco Rosales, Victor Cebotari, Franziska Gassmann.

**Writing – review & editing:** Stephan Dietrich, Aline Meysonnat, Francisco Rosales, Victor Cebotari, Franziska Gassmann.

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
