## [Decision Letter · Decision Letter 0]

1 Dec 2021

PONE-D-21-24301Economic Development, Weather Shocks and Child Marriage in South Asia: a Machine Learning ApproachPLOS ONE

Dear Dr. Dietrich,

Thank you for submitting your manuscript to PLOS ONE. After careful consideration, we feel that it has merit but does not fully meet PLOS ONE’s publication criteria as it currently stands. Therefore, we invite you to submit a revised version of the manuscript that addresses the points raised during the review process.

We look forward to receiving your revised manuscript.

Kind regards,

Santosh Kumar

Academic Editor

PLOS ONE

https://journals.plos.org/plosone/s/file?id=ba62/PLOSOne_formatting_sample_title_authors_affiliations.pdf"

3.  We note that [Figures 2 and 5] in your submission contain [map/satellite] images which may be copyrighted. All PLOS content is published under the Creative Commons Attribution License (CC BY 4.0), which means that the manuscript, images, and Supporting Information files will be freely available online, and any third party is permitted to access, download, copy, distribute, and use these materials in any way, even commercially, with proper attribution. For these reasons, we cannot publish previously copyrighted maps or satellite images created using proprietary data, such as Google software (Google Maps, Street View, and Earth). For more information, see our copyright guidelines: http://journals.plos.org/plosone/s/licenses-and-copyright.

 a. You may seek permission from the original copyright holder of Figures 2 and 5 to publish the content specifically under the CC BY 4.0 license. 

Reviewers' comments:

Reviewer's Responses to Questions

**Comments to the Author**

1. Is the manuscript technically sound, and do the data support the conclusions?

Reviewer #1: Yes

Reviewer #2: Partly

2. Has the statistical analysis been performed appropriately and rigorously? 

Reviewer #1: Yes

Reviewer #2: No

3. Have the authors made all data underlying the findings in their manuscript fully available?

Reviewer #1: Yes

Reviewer #2: Yes

4. Is the manuscript presented in an intelligible fashion and written in standard English?

Reviewer #1: Yes

Reviewer #2: Yes

5. Review Comments to the Author

Reviewer #1: Referee report “Economic Development, Weather Shocks and Child Marriage in South Asia:

a Machine Learning Approach”

The paper develops a prediction model for child marriage that relies on regional and local inputs such

as droughts, floods, population growth and nightlight data. While there is now a substantial research

at the microlevel testing the eterminants of child marriage, studies investigating macro-economic

factors contributing to child marriage and models that predict where child marriage cases are most

likely to occur remains limited. Thus, I think the paper is providing an important contribution to the

literature on the topic.

I have few comments/suggestion:

• The analysis focuses on the age bracket 20-24 and do robustness checks for the ages 18-22.

Given that they are using retrospective questions on a women age of marriage, it is not clear

to me this sample restriction. Can they use the entire sample of women 18-49? Please clarify.

• Figure 1 is very interesting. Can the authors provide a more in debt discussion relative to the

steady decline in child marriage rate in Bangladesh that until few years ago was the country

with the highest prevalence?

• The authors mentioned that, because of the practice of virocality, common in South Asia

(where a woman goes to live with the family of her groom after marriage), household

information of married women in the DHS characterize the groom’s household but not the

original household of the bride. Can’t they use information of unmarried women living in the

same cluster of the married women in the sample to make predictions?

• Another paper looking at the effect of rainfall shock on the age of marriage worth to mention

is Corno, Voena “Selling daughters: Child Marriage, Income Shocks and the Bride Price

Tradition” R&R at the JDE.

Reviewer #2: This paper builds predictive models for child-marriage in four South Asian countries using predominantly geographical and macroeconomic characteristics using Gradient Boosted Trees – a Machine Learning method. The authors find indicators of regional economic activity, income shocks and regional income levels to be common strongest predictors for child marriage across the four countries. The paper adopts an interesting and innovative approach towards using macroeconomic indicators for predicting child marriage and contributes to our understanding of how regional factors can affect social outcomes. However having read the paper very carefully, I have three major concerns about this study.

1. The idea of using regional and macro-economic indicators to predict micro-economic outcomes (child marriage) is interesting and innovative. However, with the inclusion of spatially correlated features such as night lights and droughts, the model violates the fundamental assumption of independently and identically distributed observations needed in standard Machine Learning algorithms. Machine Learning models do not have many assumptions over the distribution of the data, but it does rely on this particular assumption. And in this case spatial correlation is hardcoded in the data through the inclusion of spatially correlated covariates. The way around is the use of Machine Learning techniques appropriate for spatial data.

2. Child marriage in survey data is a rare-event – an issue the authors rightly point out. Choosing the appropriate metric for assessing prediction accuracy becomes very important here. Recall, which measures how well the model predicts the incidences of child marriage is an important metric for predicting rare-events. The authors rightly focus on it. However, Precision which measures the probability of the model’s predictions being actually correct is an equally important metric that deserves focus. The models in the paper achieve a high recall at the cost of low precision. The best model has a recall of 0.90 (implying that it accurately identifies 90% of all child marriages) and a precision of 0.25 (implying that out of the predicted child marriages by the model only 25% are indeed true child marriages). Thus, even the best model is highly overpredicting child marriages (Table 7 shows the high incidence of false positives) and as such is useless for basing any economic decisions.

A better metric is the Precision-Recall Curve or Area under Precision-Recall Curve which measures precision and recall along varying probability thresholds. Precision-Recall Curves balance the model’s tendency to underpredict and overpredict rare-events. See Brahma and Mukherjee (2021) for an illustration of using Area under Precision-Recall Curve for predicting on imbalanced data.

3. Relying only on one Machine Learning algorithm is not a wise strategy since the accuracy of models depend crucially on its ability to uncover the true underlying relationship. Another algorithm, preferably a linear model should be run to establish a benchmark.

6. PLOS authors have the option to publish the peer review history of their article (what does this mean?). If published, this will include your full peer review and any attached files.

Reviewer #1: No

Reviewer #2: No

---

## [Author Response · Author response to Decision Letter 0]

31 Mar 2022

REPLY TO COMMENTS FROM REVIEWER #1 

Manuscript: “Economic Development, Weather Shocks and Child Marriage in South Asia: a Machine Learning Approach”

Ms. Ref. No.: PONE-D-21-24301

***

Referee report “Economic Development, Weather Shocks and Child Marriage in South Asia: a Machine Learning Approach” The paper develops a prediction model for child marriage that relies on regional and local inputs such as droughts, floods, population growth and nightlight data. While there is now substantial research at the microlevel testing the determinants of child marriage, studies investigating macro-economic factors contributing to child marriage and models that predict where child marriage cases are most likely to occur remains limited. Thus, I think the paper is providing an important contribution to the literature on the topic. I have few comments/suggestion:

#1 The analysis focuses on the age bracket 20-24 and do robustness checks for the ages 18-22. Given that they are using retrospective questions on a women age of marriage, it is not clear to me this sample restriction. Can they use the entire sample of women 18-49? Please clarify.

R: Thank you for the comment. In the revised manuscript we justify the age bracket choice more clearly and detail why going further back in time is not possible with the data. By focusing on the age bracket from 20 to 24 in the main analysis we follow the standard in the child marriage literature as it reduces the risk of misreports of illicit child marriage practices which tends to be high among women who are underage at the time of the interview (28).The convention of using the retrospective view covering women aged 20-24 at the time of the interview has been set to avoid missing those women who marry after the interview but still before the age of 18 and to avoid overestimating child-marriage by counting girls aged 15-19 who are married at any given time but are 18 or 19, and thus not defined as children as per the international convention (81). In contrast, the full retroactive sample 18- to 49-year-old women who married underage, might go too far back in time such that the regional economic measures are not available. For example, the earliest interviews in our data are from 1999. A woman aged 49 in 1999 would have been underage until the year 1968. However, the nighttime light imagery we use is only available from 1992 through 2013, which is why we cannot consider older cohorts in the model. Besides this data constraint, we also have little information on events that may have happened between the interview and the time women were underage. To reduce the time span since the marriage occurred, we present results for the sample of 18–22-year-old women as a robustness test. This is clarified on page 10 in the revised manuscript. 

#2 Figure 1 is very interesting. Can the authors provide a more in debt discussion relative to the steady decline in child marriage rate in Bangladesh that until few years ago was the country with the highest prevalence?

R: The decline in child marriage rates in Bangladesh has been associated with the increasing government and non-government awareness campaigns around the issue and increasing education rates and employment opportunities for women (38, 85). Yet, the decline in the recent decade in Bangladesh has slowed. We added additional information and a reference that discusses this trend in the revised manuscript (see page 16).

#3 The authors mentioned that, because of the practice of virocality, common in South Asia (where a woman goes to live with the family of her groom after marriage), household information of married women in the DHS characterize the groom’s household but not the original household of the bride. Can’t they use information of unmarried women living in the same cluster of the married women in the sample to make predictions?

R: Thank you for the suggestion. To some extent this is what we are doing except that we aggregate at a higher level. For example, we aggregate DHS measures of women’s household decision making power, demographics, or women employment at the region level to approximate regional norms, practices, and economic characteristics (see variables of Table 2). We do not aggregate to the cluster level because the DHS data is only consistently representative down to the admin 1 level in all considered countries but not necessarily at the more granular cluster level. To avoid non-representative aggregates, we use the lowest level of aggregation for which the data is consistently representative (admin 1 regions). Another concern with more granular cluster aggregates is that information on non-married women in the cluster of origin of the brides might not give an accurate picture of the characteristics of households that led to the early marriage of girls. 

#4 Another paper looking at the effect of rainfall shock on the age of marriage worth to mention is Corno, Voena “Selling daughters: Child Marriage, Income Shocks and the Bride Price Tradition” R&R at the JDE.

R: Thank you for the reference. We have added the reference in the manuscript.

 

REPLY TO COMMENTS FROM REVIEWER #2

Manuscript: “Economic Development, Weather Shocks and Child Marriage in South Asia: a Machine Learning Approach”

Ms. Ref. No.: PONE-D-21-24301

***

This paper builds predictive models for child-marriage in four South Asian countries using predominantly geographical and macroeconomic characteristics using Gradient Boosted Trees – a Machine Learning method. The authors find indicators of regional economic activity, income shocks and regional income levels to be common strongest predictors for child marriage across the four countries. The paper adopts an interesting and innovative approach towards using macroeconomic indicators for predicting child marriage and contributes to our understanding of how regional factors can affect social outcomes. However, having read the paper very carefully, I have three major concerns about this study.

#1. The idea of using regional and macro-economic indicators to predict micro-economic outcomes (child marriage) is interesting and innovative. However, with the inclusion of spatially correlated features such as night lights and droughts, the model violates the fundamental assumption of independently and identically distributed observations needed in standard Machine Learning algorithms. Machine Learning models do not have many assumptions over the distribution of the data, but it does rely on this particular assumption. And in this case spatial correlation is hardcoded in the data through the inclusion of spatially correlated covariates. The way around is the use of Machine Learning techniques appropriate for spatial data.

R: Thank you for the comment that helped us think more deeply about the model. We carefully considered problems that may arise from the inclusion of spatially correlated features in the model. It is indeed true that the weather features are spatially correlated. As we are working with data collected from households situated in randomly selected clusters, the next closest cluster may differ greatly with respect to contextual factors like urbanicity, livelihoods, wealth etc., which results in much less spatial correlation in other features such as nighttime lights or the target variable child marriage. However, more than spatial correlation of single features, we regard possible spatial correlation of model errors as the main threat to the model validity (chapter 7 of Hastie et al., 2009 was insightful). 

As part of the revision process, we implemented several measures to reduce such concerns. First, we included latitude and longitude as additional features in the model to capture location specific influences. The results remained very similar and had no noteworthy influence on model performance (see Table 4 – S1 in the revised manuscript). Secondly, as a robustness check we used spatially blocked partitioning to validate the model as commonly applied in machine learning applications with spatial data (72). Validating the model with data from girls of randomly selected survey clusters that were not included in the training data lead to very similar results which reduces concerns that conditional on the input features model errors may not be i.i.d. (see Table S3). Thirdly, we inspected the resulting noise structure for possible spatial patterns. We computed the spatial correlation of cluster-level error rates using Moran’s I. With a power function of degree 1 as weighting matrix and approximated bilateral distances, the spatial correlation coefficients are 0.007 (Pakistan), 0.04 (Nepal), 0.03 (Pakistan), 0.08 (India), respectively, and thus do not point at weighty spatial correlations of errors. These additional tests make us confident that the inclusion of spatially correlated features does not bias our findings.

Hastie, T., Tibshirani, R., Friedman, J. H., & Friedman, J. H. (2009). The elements of statistical learning: data mining, inference, and prediction (Vol. 2, pp. 1-758). New York: springer.

#2. Child marriage in survey data is a rare-event – an issue the authors rightly point out. Choosing the appropriate metric for assessing prediction accuracy becomes very important here. Recall, which measures how well the model predicts the incidences of child marriage is an important metric for predicting rare-events. The authors rightly focus on it. However, Precision which measures the probability of the model’s predictions being actually correct is an equally important metric that deserves focus. The models in the paper achieve a high recall at the cost of low precision. The best model has a recall of 0.90 (implying that it accurately identifies 90% of all child marriages) and a precision of 0.25 (implying that out of the predicted child marriages by the model only 25% are indeed true child marriages). Thus, even the best model is highly overpredicting child marriages (Table 7 shows the high incidence of false positives) and as such is useless for basing any economic decisions.

A better metric is the Precision-Recall Curve or Area under Precision-Recall Curve which measures precision and recall along varying probability thresholds. Precision-Recall Curves balance the model’s tendency to underpredict and overpredict rare-events. See Brahma and Mukherjee (2021) for an illustration of using Area under Precision-Recall Curve for predicting on imbalanced data.

R: Thank you for the comment and reference. It is true that the choice of the performance metric can have important implications for model selection, which in our case leads to considerable overprediction of child marriage cases. We also acknowledge that the motivation for using recall in the main analysis was insufficiently discussed in the original manuscript. In the revised manuscript we motivate the choice and implications more carefully and present the results using the area under the precision recall curve as recommended. 

The main objective of the paper is to explore the potential of algorithms to support child marriage programing. Most existing anti-child marriage programs are not targeted in a narrower sense and “treatments” are assigned widely i.e., the vast majority of “treated” girls will not get married underage. For a previous project we collaborated with UN institutions in the region, where a preference was stated for algorithms that detect true child marriage cases even if this comes at the expense of limited precision. It is certainly true that our model predicts many false positives but compared to current practices we believe that this still poses a substantial improvement. By using recall as performance metric, we want to train an algorithm that ensures that the large group of “treated” girls contains as many true cases as possible. We discuss the model objectives and the performance metric implications more clearly in several parts of the revised manuscript (see page 3, 20 and 30). Based on our previous work in the field, we believe that recall is a suitable metric for this application, but ultimately the choice of the metric depends on policy preferences and the social costs societies/policies attach to child marriage cases. In the revised manuscript, we also present the results using PRC in the Supporting Information section (which leads to similar results) and show the Precision-Recall curve to illustrate the trade-off between both metrics (see Table S4 and Figure S4). 

#3. Relying only on one Machine Learning algorithm is not a wise strategy since the accuracy of models depend crucially on its ability to uncover the true underlying relationship. Another algorithm, preferably a linear model should be run to establish a benchmark.

R: Thank you for the comment. We tested several different estimators and focused the discussion on the best performing model. In the revised manuscript we now also present the results of the starting point of our analysis, a simple logistic regression, as a benchmark (see Table S2).

---

## [Decision Letter · Decision Letter 1]

30 Jun 2022

Economic Development, Weather Shocks and Child Marriage in South Asia: A Machine Learning Approach

PONE-D-21-24301R1

Dear Dr. Dietrich,

We’re pleased to inform you that your manuscript has been judged scientifically suitable for publication and will be formally accepted for publication once it meets all outstanding technical requirements.

Please add a reference suggested by R1 as that is an important paper in the child marriage literature in the final version. "R1 says- The authors have adequately addressed all my comments. They did not however missed to include the paper I was mentioning in my previous report, showing the relationship between child marriage and income shocks in Tanzania. I would suggest to include it in the final version of the manuscript."

Kind regards,

Santosh Kumar

Academic Editor

PLOS ONE

**Santosh Kumar**

Associate Professor of Economics

Department of Economics and International Business

College of Business Administration

Sam Houston State University

1803 Ave I, Huntsville, Texas 77341-2056, USA

**P**: 001 (936) 294 2416; **F**: 001 (936) 294 3488

**Email**: skumar@shsu.edu

Academic Editor, PLOS Global Health

Research Fellow, Global Labor Organization (GLO)

Research Fellow, Institute for Labor Organization (IZA)

**Webpage**: https://sites.google.com/site/santoshkumar2987/

Additional Editor Comments (optional):

Reviewers' comments:

Reviewer's Responses to Questions

**Comments to the Author**

1. If the authors have adequately addressed your comments raised in a previous round of review and you feel that this manuscript is now acceptable for publication, you may indicate that here to bypass the “Comments to the Author” section, enter your conflict of interest statement in the “Confidential to Editor” section, and submit your "Accept" recommendation.

Reviewer #1: (No Response)

Reviewer #2: All comments have been addressed

2. Is the manuscript technically sound, and do the data support the conclusions?

Reviewer #1: Yes

Reviewer #2: Yes

3. Has the statistical analysis been performed appropriately and rigorously? 

Reviewer #1: Yes

Reviewer #2: Yes

4. Have the authors made all data underlying the findings in their manuscript fully available?

Reviewer #1: Yes

Reviewer #2: Yes

5. Is the manuscript presented in an intelligible fashion and written in standard English?

Reviewer #1: Yes

Reviewer #2: Yes

6. Review Comments to the Author

Reviewer #1: The authors have adequately addressed all my comments. They did not however missed to include the paper I was mentioning in my previous report, showing the relationship between child marriage and income shocks in Tanzania. I would suggest to include it in the final version of the manuscript.

Reviewer #2: (No Response)

7. PLOS authors have the option to publish the peer review history of their article (what does this mean?). If published, this will include your full peer review and any attached files.

Reviewer #1: No

Reviewer #2: No

---

## [Editor Report · Acceptance letter]

29 Jul 2022

PONE-D-21-24301R1 

 Economic Development, Weather Shocks and Child Marriage in South Asia: A Machine Learning Approach 

Dear Dr. Dietrich:

I'm pleased to inform you that your manuscript has been deemed suitable for publication in PLOS ONE. Congratulations! Your manuscript is now with our production department. 

Kind regards, 

on behalf of

Dr. Santosh Kumar 

Academic Editor

PLOS ONE